# Shape- and Element-Sensitive Reconstruction of Periodic Nanostructures with Grazing Incidence X-ray Fluorescence Analysis and Machine Learning

**DOI:** 10.3390/nano11071647

**Published:** 2021-06-23

**Authors:** Anna Andrle, Philipp Hönicke, Grzegorz Gwalt, Philipp-Immanuel Schneider, Yves Kayser, Frank Siewert, Victor Soltwisch

**Affiliations:** 1Physikalisch-Technische Bundesanstalt (PTB), Abbestr. 2-12, 10587 Berlin, Germany; philipp.hoenicke@ptb.de (P.H.); Yves.Kayser@ptb.de (Y.K.); Victor.Soltwisch@ptb.de (V.S.); 2Helmholtz Zentrum Berlin für Materialien und Energie (HZB), Department Optics and Beamlines, Albert-Einstein-Str. 15, 12489 Berlin, Germany; grzegorz.gwalt@helmholtz-berlin.de (G.G.); frank.siewert@helmholtz-berlin.de (F.S.); 3JCMwave GmbH, Bolivarallee 22, 14050 Berlin, Germany; philipp.schneider@jcmwave.com; 4Zuse Institute Berlin, Takustrasse 7, 14195 Berlin, Germany

**Keywords:** GIXRF, Bayesian optimization, periodic nanostructure

## Abstract

The characterization of nanostructured surfaces with sensitivity in the sub-nm range is of high importance for the development of current and next-generation integrated electronic circuits. Modern transistor architectures for, e.g., FinFETs are realized by lithographic fabrication of complex, well-ordered nanostructures. Recently, a novel characterization technique based on X-ray fluorescence measurements in grazing incidence geometry was proposed for such applications. This technique uses the X-ray standing wave field, arising from an interference between incident and the reflected radiation, as a nanoscale sensor for the dimensional and compositional parameters of the nanostructure. The element sensitivity of the X-ray fluorescence technique allows for a reconstruction of the spatial element distribution using a finite element method. Due to a high computational time, intelligent optimization methods employing machine learning algorithms are essential for timely provision of results. Here, a sampling of the probability distributions by Bayesian optimization is not only fast, but it also provides an initial estimate of the parameter uncertainties and sensitivities. The high sensitivity of the method requires a precise knowledge of the material parameters in the modeling of the dimensional shape provided that some physical properties of the material are known or determined beforehand. The unknown optical constants were extracted from an unstructured but otherwise identical layer system by means of soft X-ray reflectometry. The spatial distribution profiles of the different elements contained in the grating structure were compared to scanning electron and atomic force microscopy and the influence of carbon surface contamination on the modeling results were discussed. This novel approach enables the element sensitive and destruction-free characterization of nanostructures made of silicon nitride and silicon oxide with sub-nm resolution.

## 1. Introduction

Since nanotechnology and thus nanostructures of different kind are relevant in many areas of science and technology, metrology techniques that can support design, research, and fabrication of such nanostructures are of high importance. Especially in the semiconductor industry, which is probably the most popular field of application for nanotechnology as well as a strong driver for research in this field, complex 2D and 3D nanostructures with feature sizes in the single-digit nanometer regime [1,2] are employed in order to keep Moore’s law [3] alive. The performance of these nanostructures crucially depends on a well-controlled fabrication, both in terms of targeted dimensional parameters and 3D element compositions (e.g., dopant distributions). Thus, there is a strong need for metrology techniques that allow us to characterize these parameters with sufficient sensitivity [4].

Typical analytical or dimensional techniques used in this context are scanning and transmission electron microscopy (SEM, TEM) [5,6], atomic force microscopy (AFM) [7], and techniques that also address elemental distributions such as secondary ion mass spectroscopy (SIMS) [8], atom probe tomography (APT) [9], and energy-dispersive X-ray spectroscopy (EDX) combined with scanning electron microscopy (STEM) [10]. All of these techniques have different advantages and disadvantages regarding sample preparation and consumption, achievable spatial resolution, required duration, and other experimental parameters (e.g., tip sizes). Optical metrology based on light–matter interaction has a significant advantage in terms of measurement speed (with respect to slow techniques, e.g., APT and STEM) and the ability to statistically measure large areas in contrast to scanning techniques. Optical reflectometry, also known as optical critical dimension (OCD) [11] metrology, is still used and is continuously improved despite the resolution limits that have been reached. In order to keep pace with shrinking structures, intensive research is being carried out to reduce the wavelength of the employed radiation and to increase the sensitivity via the dispersion of the periodic nanostructured surface. This is called deep ultraviolet (DUV) or extreme ultraviolet (EUV) scatterometry and can be extended into the X-ray spectral range. In X-ray scattering techniques, a distinction is made between measurements in transmission, also known as critical dimension small angle X-ray scattering (CDSAXS) [12], and reflection mode, known as grazing incidence small angle X-ray scattering (GISAXS) [13,14]. Both techniques have already shown that they allow for the dimensional reconstruction of nanostructures with an uncertainty in the sub-nm range [15,16]. CDSAXS can require a special sample thinning. Usually, these techniques also employ rather high photon energy X-rays, which limits the optical contrast between different materials within the investigated nanostructures.

In this paper, we employ the grazing incidence X-ray fluorescence analysis (GIXRF) [17] method and soft X-rays (<1 keV) instead to analyze the nanostructured surface. This nondestructive technique allows reconstructing the composition of the sample in terms of both their dimensional properties as well as the distribution of different elements, since the emitted X-ray fluorescence (XRF) is characteristic for each element. GIXRF uses the X-ray standing wave (XSW) field [18,19], which results from the interference between the incident and the reflected X-ray beam as a nanoscale sensor. The intensity modulation inside the XSW field significantly influences the X-ray fluorescence intensity of an element depending on its spatial position within the electromagnetic field distribution. Recent studies have shown the potential of the GIXRF technique for the dimensional and compositional nanometrology of periodic 2D [20] and 3D [21,22] nanostructures.

In this work, we further develop this approach toward being a reliable metrology technique by employing a combined element sensitive reconstruction for the fluorescence signals of oxygen, nitrogen, and, in parts, also carbon from within a silicon nitride grating structure. Artificial intelligence and machine learning (ML) techniques are studied intensively for a wide range of applications [23,24,25]. These techniques can help to improve the optimization of different materials. The integration of ML, such as the Bayesian optimization [26] algorithm based on Gaussian processes in combination with a finite-element Maxwell solver [27] allows to control the computational modeling effort and to derive first estimates of the uncertainties of the parameterized nanostructure. For an even increased reliability, we are experimentally determining the optical constants of the employed materials instead of using tabulated data, as these are often not realistic for nanolayers and nanostructures [28]. Finally, we compare and validate the results against AFM and SEM cross sections.

## 2. Materials and Methods

### 2.1. Experimental

As an example of two-dimensional nanostructures, a lithographically patterned silicon nitride grating on a silicon substrate was investigated. It was manufactured by means of electron beam lithography (EBL) at the Helmholtz-Zentrum Berlin. The nominal pitch of the grating is p=100 nm, the nominal height is h=90 nm, and the nominal line width of the sample is w=50 nm. For the manufacturing of the gratings, a silicon substrate with a 90 nm-thick Si${}_{3}$N${}_{4}$ layer was used. ZEP520A, a positive resist (organic polymer), was spin-coated on the substrate and developed with a Vistec EBPG5000+ e-beam writer, operated with an electron acceleration voltage of 100 kV. The grating was etched via reactive ion etching using CHF${}_{3}$ and to remove the remaining resist an oxygen plasma treatment was applied. A sketch of the cross-section is shown in Figure 1a). The total structured area of the grating was 1 mm × 15 mm and the sample area outside the patterned region consists of the originally deposited Si${}_{3}$N${}_{4}$ layer. Directly after fabrication, cross-section SEM images have been recorded on a sister sample.

We performed GIXRF and X-ray reflectometry (XRR) measurements in PTB’s laboratory [29] at the BESSY II electron storage ring using the plane-grating monochromator (PGM) beamline [30] for undulator radiation. The sample was mounted in an ultrahigh-vacuum (UHV) measurement chamber [31], where a nine-axis manipulator allows for an accurate sample alignment with respect to the direction of incident X-ray beam. The incidence angle θ is defined as the angle between the X-ray beam and the sample surface. The azimuthal angle φ is defined as the angle between the incident beam and a plane, which is normal to the sample surface and parallel to the direction of the grating lines such that $\varphi =0^{\circ }$ is defined as the orientation parallel to the plane of incidence. Both sample rotation axes can be aligned with an uncertainty below 0.01${}^{\circ }$.

As we have employed radiometrically calibrated X-ray fluorescence (XRF) instrumentation, we can perform reference-free GIXRF [32] and gain a quantitative access to the elemental mass depositions present on the sample [33]. At each angular position for θ and φ, a fluorescence spectrum is recorded with a calibrated silicon drift detector (SDD) [34] and the incident photon flux is monitored by means of a calibrated photodiode. The GIXRF-measurements were performed at an incident photon energy of $E_{i}=680$ eV, allowing for the excitation of N-Kα as well O-Kα fluorescence radiation, which mainly originates from the surface oxide layer on the grating structure.

In addition, we have performed XRR experiments on the nonstructured Si${}_{3}$N${}_{4}$ layer next to the grating at the same photon energy ($E_{i}=680$ eV). From this, we can determine the optical constants of the SiO${}_{2}$ layers and the Si${}_{3}$N${}_{4}$ layer, which are expected to be more reliable in the soft X-ray spectral range than using only tabulated data as in [35].

For an independent validation of the dimensional GIXRF reconstruction results, additional AFM measurements were performed with an Nanosurf Nanite 25 × 25. The sample was measured under tapping mode condition and a standard pyramidal shaped silicon probe with a tip radius <10 was used, as it is commonly applied for the inspection of diffraction gratings [36]. The AFM probe is characterized by a resonance frequency of 190 Hz and force constant of 48 N/m. The inspected sample area was 500 × 500 ${\mathrm{nm}}^{2}$ in size covering about five grating lines.

### 2.2. Simulation and Optimization of Fluorescence Intensities

The GIXRF signals are directly related to the XSW field intensity distribution, which, besides the incident photon energy and the incidence and azimuthal angles, depends on the shape and material composition of the illuminated nanostructured surface. To reconstruct these sample features from the experimental data, we applied a finite-element-method (FEM)-based forward calculation of the XSW and optimized the structural parameters to reproduce the experimental data.

In Figure 1, the basic principle of the FEM procedure is shown. The mesh grid and the model parameters are displayed in Figure 1a. The FEM solver calculates the electric near-field for a given structure (Figure 1b). The amount of fluorescence photons generated at a given coordinate depends on the local electric near-field intensity (E(x,y)) and the compositional and fundamental parameters of the respective material (mass fraction of the fluorescent element in the material $W_{k}$, photoionization cross-section for the incident photon energy $\tau (E_{i})$, and the fluorescence yield $\omega_{k}$, taken from databases [37] or dedicated experiments [38]). These photons can be reabsorbed on the path through the sample toward the detector (Figure 1b) with a probability depending on the materials mass attenuation coefficient $\mu (E_{f})$ for the fluorescence photon energy $E_{f}$, the density ρ of the material, and the distance to the surface of the nanostructure $y_{dis}\left(x,y\right)$. Eventually, the fluorescence photon will be detected with a given detection efficiency $\epsilon (E_{f})$ if it is arriving within the effective solid angle of detection Ω. The overall emitted fluorescence intensity also depends on the incident photon flux $N_{0}$.

Thus, the measured emitted fluorescence intensity $\Phi (\theta ,\varphi ,E_{i})$ (derived from the detected count rate $F(\theta ,\varphi ,E_{i})$) can be modeled using the integration over the full area of the fluorescent material of the nanostructure as described by the modified Sherman equation [20,22,39]:(1)$$\Phi \left(\theta ,\varphi ,E_{i}\right)=\underset{\begin{array}{c} I_{exp} \end{array}}{\underset{︸}{\frac{4\pi \sin\theta }{\Omega }\frac{F(\theta ,\varphi ,E_{i})}{N_{0}\epsilon \left(E_{f}\right)}}}=\underset{\begin{array}{c} I_{model} \end{array}}{\underset{︸}{\frac{W_{k}\rho \tau \left(E_{i}\right)\omega_{k}}{\sum dx}\;\cdot \;\sum_{x}\sum_{y}{\left|E\left(x,y\right)\right|}^{2}\;\cdot \;\exp\left[-\rho \mu \left(E_{f}\right)y_{dis}\left(x,y\right)\right]dxdy}}.$$

Equation (Equation 1) is applied for nitrogen and oxygen fluorescence and has been implemented directly in the Maxwell solver to eliminate errors due to conversion to a regular Cartesian grid. The calculated emitted fluorescence intensities $I_{model}$ are then compared against the experimental value $I_{exp}$. An optimal set of the model parameters can then be determined through use of a global optimization algorithm such as Bayesian optimization (BO) [26,27]. BO uses a stochastic model, a Gaussian process, of an unknown objective function to be minimized in order to determine promising parameter values [26,40]. The mathematical background of Bayesian optimization can be found in [26]. The expected improvement EI is crucial to find the best parameters to evaluate the function and identify the global minimum. The EI of an unknown function can be calculated with the Gaussian process. The next sampling point is where the EI is at its maximum
(2)$$x_{n+1}=\mathrm{argmax}\left(EI_{n}\left(x\right)\right).$$

The EI is high where we have not evaluated the function or found values close to a minimum of the function. In a previous work [27,41], it was shown that the BO performs much better than other metaheuristic optimization approaches with respect to the computing time needed to find the global minimum. Since BO considers all previous function evaluations, it can be more efficient than other metaheuristic global optimization strategies [27] and local optimization strategies [42]. This is a crucial benefit here, as one model calculation takes several minutes on a standard desktop computer. For a detailed benchmark study of the different optimizer methods in comparison to BO, see [27,41]. For our problem, the BO gives good results in a reasonable amount of time [41]. Here, we use an implementation of BO that is part of the JCMsuite software package [43].

The error function $\chi^{2}$
(3)$$\chi^{2}\left(\vec{gp}\right)=\sum_{\theta ,\varphi }\frac{{\left(I_{exp}\left(\theta ,\varphi \right)-I_{model}\left(\vec{gp},\theta ,\varphi \right)\right)}^{2}}{\sigma_{N}^{2}\left(\theta ,\varphi \right)}$$
was minimized with respect to the different model parameters of $\vec{gp}$ described earlier (see Figure 1a and model errors for the nitrogen $\epsilon_{N}$ and the oxygen fluorescence signals $\epsilon_{O}$). These model error parameters are introduced to consider potential errors, e.g., the uncertainty of the employed atomic fundamental parameters or the influence of a thin surface contamination layer into account. Even though material-dependent parameters such as optical constants or material densities deviate most likely from the tabulated bulk data for the grating materials (SiO${}_{2}$ and Si${}_{3}$N${}_{4}$), we do not include these as free parameters in the model. Indeed, as this would drastically increase the number of model parameters and thus prolong the necessary calculation times, we determine these parameters separately using XRR (described in the next section). $\sigma_{N}$ is the calculated experimental error consisting of an error estimation for the effective solid angle of detection $\sigma_{\Omega }\left(\theta \right)$ and the error contributions originating from counting statistics $\frac{\sqrt{F}}{F}$ for the respective fluorescence line as well as for the spectra deconvolution $\sigma_{num}$.
(4)$$\sigma_{N}{\left(\theta \right)}^{2}={\left(\frac{\sqrt{F(\theta )}}{F(\theta )}\right)}^{2}+\sigma_{\Omega }{\left(\theta \right)}^{2}+\sigma_{num}{\left(\theta \right)}^{2}$$

By inverting the Hessian matrix $H_{kj}$ [44] of the error function,
(5)$$H_{kj}=\frac{\partial^{2}\chi^{2}\left(\vec{gp}\right)}{\partial gp_{j}\partial gp_{k}}$$
at the minimum of $\chi^{2}$, it is possible to determine the confidence intervals ${\left(H\right)}^{-1}$ of the model parameters, if they are Gaussian distributed as assumed (Assumption 1).

The standard deviation or noise level at the global minimum parameter set is defined as $\mathrm{STD}=\sqrt{\frac{\chi^{2}}{\mathrm{DOF}}}$, where DOF=N−M is the difference between the number of measurement points *N* and the number of free model parameters *M*, and thus the degrees of freedom.

The model parameter confidence interval can then be calculated as
(6)$$\sigma_{\vec{gp}}=\mathrm{STD}\sqrt{\mathrm{diag}({\left(H\right)}^{-1})}$$

Based on the Gaussian process model of $\chi^{2}\left(\vec{gp}\right)$, we determine an estimate of the Hessian matrix by computing all second derivatives of the Gaussian process model of $\chi^{2}$ at the minimum and can then estimate the error covariance matrix ${\left(H\right)}^{-1}$ and the parameter confidence interval.

## 3. Results and Discussion

### 3.1. Validation of the Optical Material Parameters

From the angle-dependent measurement of reflection intensities on a layered system, information about layer thicknesses, densities, or even their optical constants can be obtained [45]. This is well known and widely used. Figure 2b shows the experimental data in comparison with the best simulation. The high frequency oscillation visible in the XRR curve is a clear indication of a multilayer system. Due to the native oxide layer of the Si substrate (Assumption 2) and the removal of the photo resist, resulting in an oxidized surface on the Si${}_{3}$N${}_{4}$ layer, which is well known due to the oxygen plasma cleaning [46], we apply a three-layer model for the XRR simulation (as shown in the inset of Figure 2a). Since the calculation of the reflectivity for a 1D layer system is several orders of magnitude faster than the FEM based 2D GIXRF modeling, statistical analysis methods of the posterior distributions can be used for a large number of parameters such as layer thickness, roughness, and optical constants. We applied the Markov chain Monte Carlo method (MCMC) [47] to determine the individual parameter uncertainties and to resolve possible interparameter correlation effects [48]. In Figure 2a, the posterior distribution determined in this procedure is shown as projections of the refractive index $n(Si_{3}N_{4})=1-\delta +i\beta$ and thickness *h* of the $Si_{3}N_{4}$ layer. The almost perfect Gaussian-like shape of the distributions, which is also present in all other parameters, allows the determination of uncertainties directly from the measurements. The relative uncertainties of the experimental data reconstructed with a linear error model (ax+b) are with (0.6±0.2)% exactly in the expected range.

From this modeling, we derived optical constants for the top $SiO_{2}$ (δ = $\left(8.73\pm 0.03\right)10^{-4}$, β = $\left(2.58\pm 0.04\right)10^{-4}$) and the $Si_{3}N_{4}$ (δ = $\left(12.59\pm 0.05\right)10^{-4}$, β = $\left(2.58\pm 0.02\right)10^{-4}$) layers. By comparing the experimentally determined optical constants with tabulated Henke data [49], one can estimate the densities of the respective materials and their deviation from the respective bulk densities. Relative densities of (0.89±0.01) and (0.79±0.01) were found for $Si_{3}N_{4}$ and $SiO_{2}$, respectively. This is in line with the already observed material density reduction discussed in [20]. The reduced densities as well as the experimental optical constants are used for the GIXRF reconstruction.

### 3.2. GIXRF Reconstruction Results

#### 3.2.1. Virtual Experiment

Before we apply the reconstruction model to real experimental data, we apply it to an artificial data set in order to test the reconstruction method and to study whether an increased incident photon energy that is capable of also exciting an oxygen fluorescence signal is beneficial. In our previous study [20], measurement data at $E_{i}=520$ eV were analyzed and an indirect sensitivity to the surface oxide layer was found, even though no fluorescence signal originating from it was used for the reconstruction. The indirect sensitivity was merely due to the attenuation behavior of the oxide layer; thus, we expect an increased sensitivity if a direct signal originating from it is also used. For this purpose, we generate artificial experimental data by calculating model curves using the reconstruction model for a given set of parameters and the two incident photon energies of $E_{i}=680$ eV and $E_{i}=520$ eV for the first time. To mimic experimental noise, we apply a Gaussian disturbance with a width of 3% (see Table 1). By reconstructing the artificial data sets, we can now analyze the influence of the increased incident photon energy on the reconstruction results and their confidence intervals without the influence of any experimental error contributions.

In Figure 3, the corresponding artificial GIXRF curves and reconstruction results are displayed. They match the artificial data well for both photon energies and characteristic features, e.g., the first local maxima are well retrieved. From the BO reconstruction, we are able to calculate the confidence intervals of the reconstructed parameters and, as shown in Table 1, they agree well with the initial parameters within the derived confidence interval.

From a comparison of the determined confidence intervals for the two different configurations (Configuration A, $E_{i}=520$ eV, only the nitrogen signal is modeled and Configuration B, $E_{i}=680$ eV, both nitrogen and oxygen signals are modeled; see Table 1), the positive influence of also considering the oxygen fluorescence signal is obvious. Especially for the height and the groove oxide thickness, the achieved confidence intervals are significantly smaller for configuration B. Nevertheless, for configuration A, the angle-dependent nitrogen fluorescence contains all relevant information about the dimensional properties of the nanostructure and even the surface oxide layer, as already pointed out in our earlier work [20], but here, we investigated it systematically with simulated data and the calculation of the confidence intervals.

#### 3.2.2. Real Experimental Data

Using the same model and methodologies, the real experimental data, measured according to configuration B, were modeled. A comparison of the two experimental data sets and the resulting modeled data is shown in Figure 4. The obtained and normalized N-Kα (a) and O-Kα (b) fluorescence intensities for different angles θ and φ are shown. The corresponding optimal parameters and confidence intervals are summarized in Table 2. Again, the confidence intervals of the structure parameters derived from the BO posterior distribution are similar to those determined in the virtual experiment.

Similar to the result from the virtual experiment, a higher sensitivity for the line width is observed as compared to the height. This is expected to be a result of the relatively large line height with respect to the achievable information depth in the soft X-ray regime. Thus, the nitrogen fluorescence radiation from the bottom of the grating line does not contribute significantly to the overall observed signal. Nevertheless, all calculated confidence intervals are in the sub-nm regime.

It should also be noted that the two modeling error parameters $\epsilon_{N}$ and $\epsilon_{O}$ deviate from unity within a range of 10%. This is the same magnitude that one would expect the relative uncertainty of the employed fluorescence production cross-sections (product of fluorescence yield and photo ionization cross section) to be in.

To validate whether the calculated confidence intervals are in a realistic magnitude, we repeated a GIXRF measurement ten times at $\varphi =0^{\circ }$ on the same sample and position and applied the reconstruction for every single measurement. We calculated the standard deviation of all model parameters and found that it is in a similar regime as the calculated confidence intervals. Parameter correlation cannot be identified from the Hessian matrix, as we expected from previous investigation based on a Bayesian inversion study [50]. Systematic errors not taken into account will increase the final uncertainties. This problem is often called model error and refers to the whole physical model or virtual experiment that is applied and is not limited to the finite element model. The next section shows that these modeling uncertainties can have a significant impact on the reconstruction parameters.

#### 3.2.3. Influence of Model Errors

In addition to oxygen and nitrogen signals in the fluorescence spectra, carbon fluorescence was also observed. A presence of carbon on the sample surface is likely as the sample is stored under normal ambient conditions. A lateral scan of the sample (Figure 5a) at a fixed incident angle $\theta =15^{\circ }$ reveals that this carbonaceous contamination is not homogeneously distributed over the patterned area. In the center along the lines of the grating area (x=−0.15 mm), a strong increase of the carbon signal and a slight increase of the oxygen signal can be observed, whereas the nitrogen signal is practically constant. This can be connected to a decrease in the reflected signal and it can be observed as a dark line along the grating in a microscope image. In an earlier benchmark study, the sample under investigation was measured in various scatterometers and electron microscopes around the world. A contamination of the grating surface by these techniques can therefore not be excluded. Here, we take advantage of this unintentionally created contamination layer to demonstrate the sensitivity and the influence of model errors when using a model without contamination layer. At each lateral position shown in the figure, we performed a GIXRF angular scan (at φ = 0${}^{\circ }$) and performed the reconstruction, including the confidence interval calculation, without considering any contamination. By comparing the different GIXRF scans (not shown here), a difference in the intensity of the fluorescence signal can be observed for the different elements as a function of the angle of incidence.

Figure 5b shows the obtained differences between the reconstructed model parameters at each position to the reference position at x=0 (position where the data shown in Figure 4 was taken). The model parameters at −0.15 mm, which is the position with maximal carbon and oxygen contamination, differ clearly with respect to less contaminated areas. The difference is larger than the calculated confidence intervals from the reconstruction, and they are also much larger than the expected lateral inhomogeneities of the sample. In addition, the nearly constant nitrogen fluorescence signal in the *X* direction clearly indicates a homogeneous overall amount of $Si_{3}N_{4}$, which is in contradiction to the larger cross-sectional area of the grating determined by the reconstruction from the GIXRF model. The reconstructed heights and widths increase by more than 4 nm.

This behavior can be explained considering the XSW near-field distribution inside the grooves. The slight increase of the oxygen due to the contamination signal can only be incorporated by increasing the oxide layer thicknesses. However, an increase of the oxide layer thickness weakens the penetrating field inside the $Si_{3}N_{4}$ and thus the emitted nitrogen fluorescence. To compensate for this, a larger grating cross section is reconstructed. One may think that the reconstruction algorithm could circumvent this by simply increasing only the groove oxide layer thickness, which does not affect the nitrogen signal so much. However, as shown in part (c) of the figure, where the angular oxygen fluorescence signal contributions from the different parts in the structure model are shown, the groove oxide has a significantly different angular behavior as compared to the oxide on the grating line surface. The features at about $1^{\circ }$ and $2.7^{\circ }$ are especially distinct. For this reason, the reconstruction algorithm must increase both in order to account for the higher oxygen signal in the contaminated area throughout the angular range.

This shows for the first time not only how sensitive the GIXRF method is but also how carefully the models have to be developed for a realistic uncertainty estimation. For a complete uncertainty analysis, however, further influences must be investigated. Effects such as roughness, alignment errors, and inhomogeneous atomic distributions within the structures may lead to larger uncertainty contributions. However, even if the model is not accurate enough, a spatially resolved reconstruction can reveal flaws of the model.

### 3.3. Comparison with SEM and AFM

For a validation of the dimensional and compositional parameters as derived from the GIXRF modeling without the thick carbon contamination, we have performed AFM measurements on the sample and also used SEM (on a witness sample). As AFM is an established method to map the surface topology of nanostructures with sensitivities down to the nm regime, we can use an AFM to verify the GIXRF reconstructed total line heights (difference between top surface and groove surface). From the AFM data, we determined a total line height of 98.4 nm. This is consistent with the GIXRF reconstructions within the confidence intervals. Here, it should be noted that the AFM measurement is only representative for the small area of the grating where it was performed whereas the GIXRF result is averaged over a much larger area due to the elongated beam footprint. Due to the interplay of the tip shape with the nanostructure, other dimensional parameters such as, e.g., line width or sidewall angle are not deduced in a straightforward manner [51].

In Figure 6 the GIXRF-reconstructed shape of the sample as well as the AFM profile, obtained by averaging two in juxtaposition located AFM line profiles, is overlayed onto the SEM cross-section. Therefore, the GIXRF line profile and the AFM were scaled to the SEM image in order to match the line pitch while keeping the aspect ratios constant. As far as the low contrast allows, the agreement with respect to line height, sidewall angle, line width, and even the oxide layer thicknesses (assuming the bright areas in the SEM to be the oxide, Assumption 3) is reasonably good. The reconstructed thicker groove oxide can also be seen in the SEM picture. The difference is greater than expected from the confidence intervals, but as stated before, the SEM image was taken from a witness sample. Therefore, an exact match is unlikely due to inaccuracy in the deposition of the Si${}_{3}$N${}_{4}$ layer and in the overall production process.

## 4. Conclusions

Here, we have demonstrated how the GIXRF-based methodology for a dimensional and compositional characterization of regular nanostructures can be enhanced with respect to the achieveable sensitivities by incorporating fluorescence signals of different elements from within the nanostructure. In addition, the incorporation of supporting experiments such as, e.g., XRR for optical constant verification and machine learning techniques such as Bayesian optimization decrease the necessary computational effort of the FEM-based reconstruction. The BO allows for intelligent and fast scanning of the parameter space as compared to other optimizer approaches.

For the reconstruction, we need to make the following assumptions. For the calculation of the confidence intervals, we need to assume that the posterior distribution of the model parameters are Gaussian distributed. For the used model, we assumed that the Si substrate is oxidized and that any contamination is negligible. For the comparison of the shape of the nanostructure, we assumed that the bright areas in the SEM image represent the oxide.

Initial steps toward determining a reliable uncertainty budget for the reconstructed parameter set were taken by deriving confidence intervals for the parameters from the Gaussian process model. We have shown how the incorporation of the oxygen signal shifts the achievable sensitivities well into the sub-nm regime. This method can also be applied to various systems as described in this paper [22]. The obtained GIXRF reconstruction results agree well with results from SEM and AFM, indicating the validity of the methodology.

In addition, we have shown that the methodology is also somewhat sensitive toward unexpected effects on the nanostructure using the example of the carbon contamination. Notably, the element sensitivity of X-ray fluorescence and the behavior of the reconstruction results indicate if unexpected effects are present on the nanostructure.

By developing more sophisticated techniques to quantify the corresponding model error influences on the final parameter uncertainties, this can be a promising technique for nanostructure characterization. In fact, by combining it with techniques such as soft X-ray scattering, one may even enhance the obtainable sensitivities and learn about important quality parameters, e.g., line roughnesses [16].

## Figures and Tables

**Figure 1 nanomaterials-11-01647-f001:**
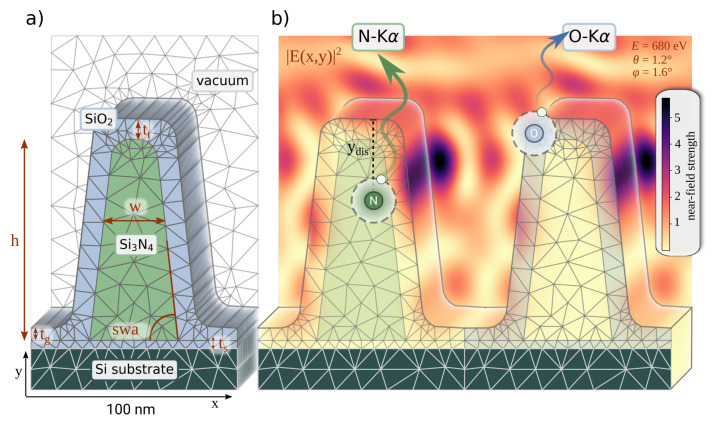
(**a**) Cross-section with the finite-element mesh grid showing the layout used for the simulation. The height *h*, the width *w*, the sidewall angle swa, and the oxide layer thicknesses in the groove $t_{g}$ and on the grating line $t_{t}$ were optimized as independent parameters. The oxide layer on the substrate $t_{s}$ was kept constant during the optimization. (**b**) The calculated electric field strength inside and outside the structure is shown for $\theta =1.2^{\circ }$ and $\varphi =1.6^{\circ }$. For the nitrogen and oxygen fluorescence, the electric field strength is integrated and the reabsorption is calculated with the distance y${}_{dis}$ the photon has to travel to leave the structure.

**Figure 2 nanomaterials-11-01647-f002:**
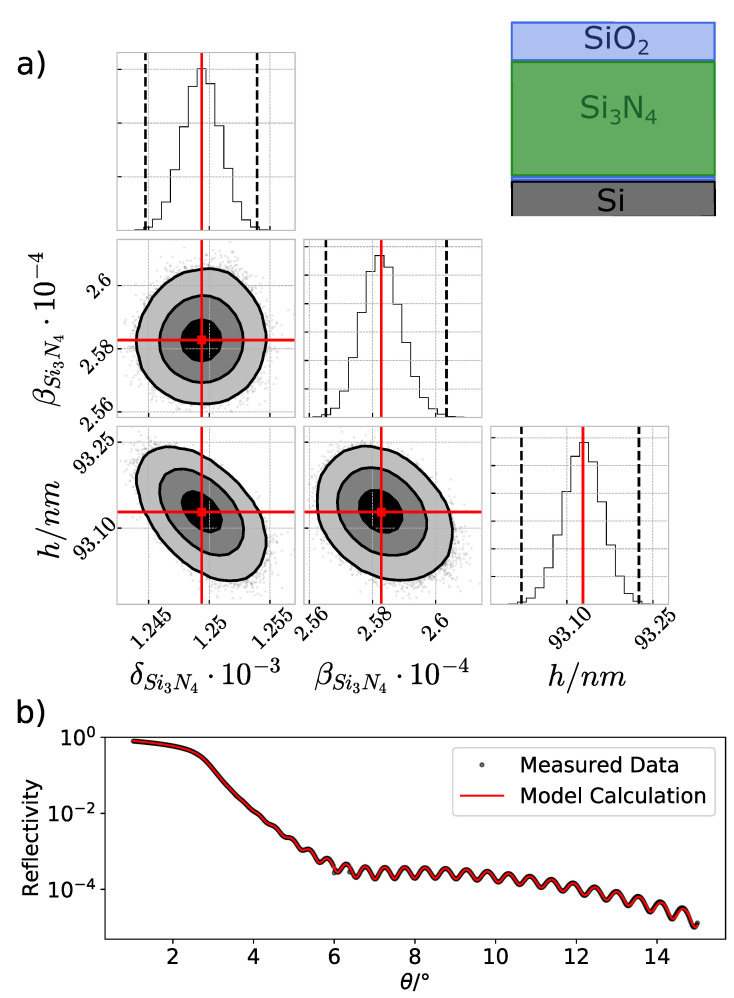
(**a**) The posterior distributions for relevant parameters $\delta_{Si_{3}N_{4}}$, $\beta_{Si_{3}N_{4}}$, and the $Si_{3}N_{4}$ thickness *h* obtained from the MCMC sampling. The red line marks the mean of the distribution and the dotted black lines in the histogram indicates a 3σ interval. In the top right corner, a sketch of the used layer stack is shown. (**b**) Comparison of the experimental data (black stars) and the model calculation (red line) as obtained by the MCMC.

**Figure 3 nanomaterials-11-01647-f003:**
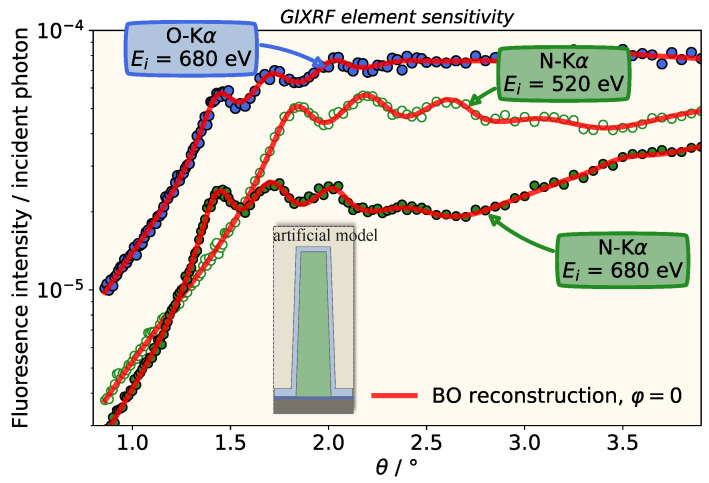
Comparison of the expected artificial disturbed simulated N-Kα (green points) or O-Kα (blue points) fluorescence intensities for the different excitation energies and the corresponding BO reconstruction. Clearly visible is how the first peak in the nitrogen signal shifts with increasing energy to smaller incident angles.

**Figure 4 nanomaterials-11-01647-f004:**
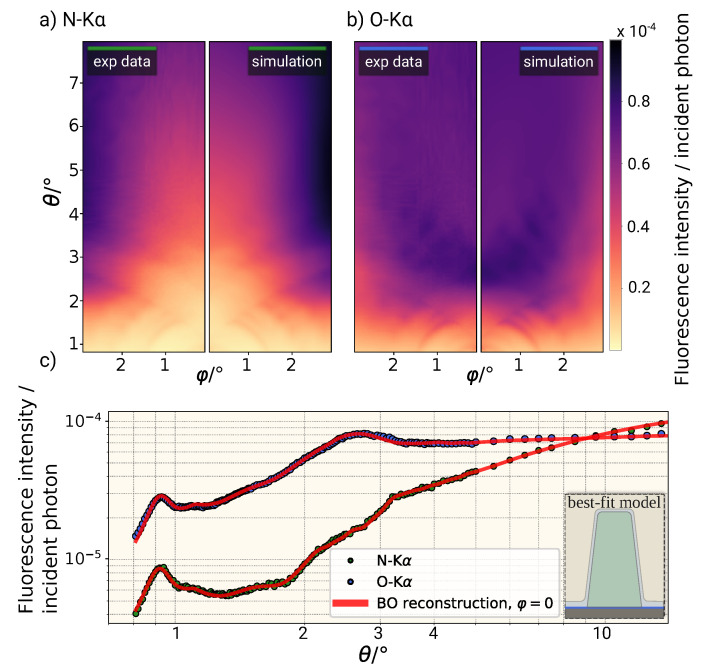
Comparison of the measured and simulated fluorescence maps for N-Kα (**a**) and O-Kα (**b**) based on the reconstructed parameter set. (**c**) Comparison of the experimental N-Kα (green) or O-Kα (blue points) fluorescence intensities for $\varphi =0^{\circ }$ to the reconstructions from the Bayesian optimization (red lines).

**Figure 5 nanomaterials-11-01647-f005:**
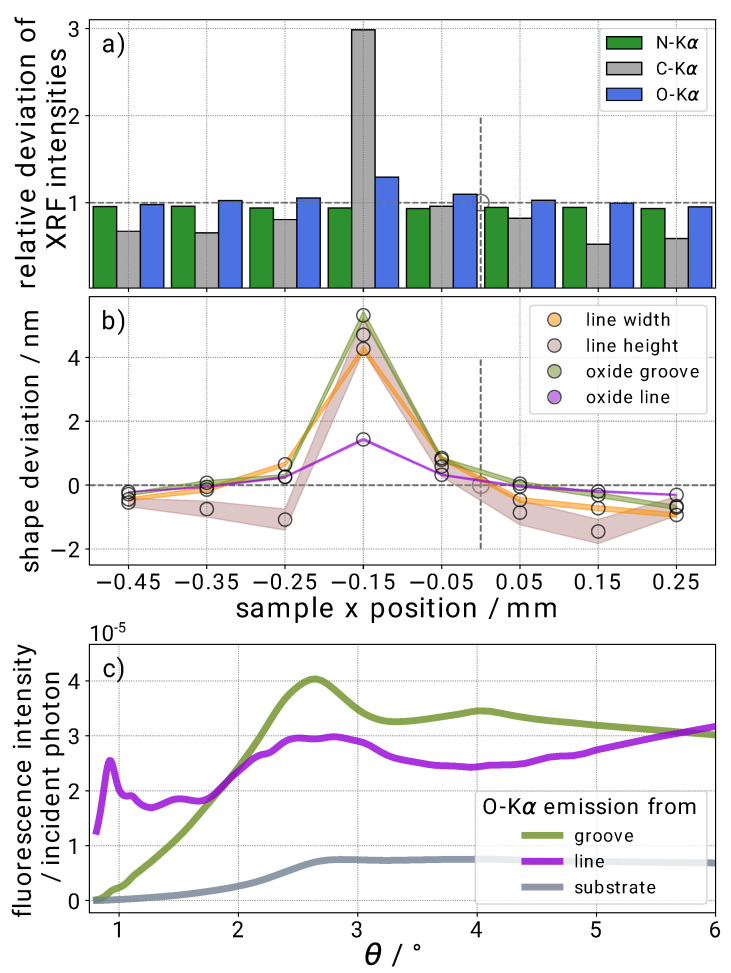
(**a**) Here, the C-Kα, N-Kα, and O-Kα fluorescence intensities at $\theta =15^{\circ }$ where no XSW needs to be considered from different positions on the sample are compared. (**b**) The reconstruction results for the width, the height, the thickness of the oxide in the groove, and the thickness of oxide around the line for the different positions are plotted. (**c**) The various contributions from the line, the groove, and the oxide layer of the silicon wafer to the total oxygen fluorescence are compared.

**Figure 6 nanomaterials-11-01647-f006:**
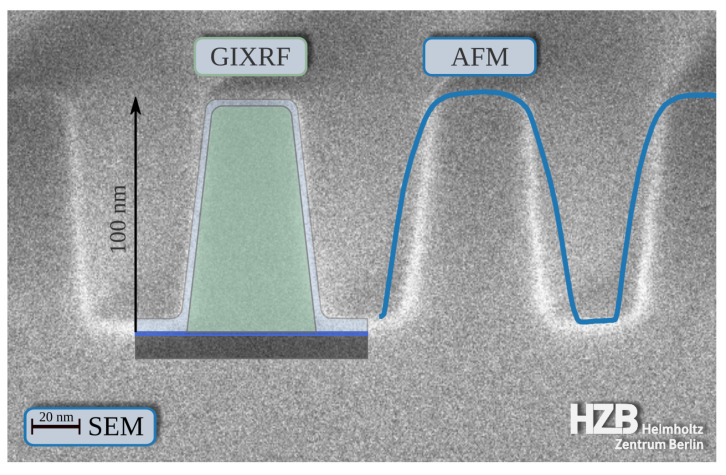
Comparison of the AFM data with the result of the GIXRF reconstruction and an SEM picture from a witness sample.

**Table 1 nanomaterials-11-01647-t001:** The values of the geometrical parameters of the synthetic data from the GIXRF BO reconstruction with the model parameter confidence intervals (CI) for $E_{i}=520$ eV (Configuration A, only the nitrogen signal is modeled) and $E_{i}=680$ eV (Configuration B, both nitrogen and oxygen signals are modeled). The pitch was set to p=50 nm. (height *h*, width *w*, sidewall angle swa, oxide layer in the groove $t_{g}$, and oxide layer on the grating line $t_{t}$).

Parameter	Intial	Config.				Ratio
Name	Value	A	${CI}_{A}$	B	${CI}_{B}$	$\frac{{CI}_{B}}{{CI}_{A}}$
h/nm	90	89.4	0.6	90.3	0.4	0.67
w/nm	25	24.87	0.07	25.08	0.06	0.85
$swa{/}^{\circ }$	88	88.1	0.1	87.9	0.1	1.0
$t_{t}/$nm	3	3.13	0.06	3.02	0.05	0.83
$t_{g}/$nm	5	5.3	0.6	5.0	0.1	0.17
$\epsilon_{N}$	1	1.02	0.01	1.00	0.01	1
$\epsilon_{O}$	1	-		0.99	0.01	

**Table 2 nanomaterials-11-01647-t002:** The values of the geometrical parameters of the nanostructures from the GIXRF BO reconstruction with the model parameter confidence intervals (one sigma). The parameters are height *h*, width *w*, sidewall angle swa, oxide layer in the groove $t_{g}$, and oxide layer on the grating line $t_{t}$.

Parameter	Reconstructed	Confidence Intervals
Name	Value	Value
h/ nm	97.5	0.5
w/nm	49.77	0.07
$swa{/}^{\circ }$nm	83.54	0.09
$t_{t}/$nm	2.84	0.03
$t_{g}/$nm	5.82	0.09
$\epsilon_{N}$	0.918	0.006
$\epsilon_{O}$	1.059	0.007

## Data Availability

The data presented in this study is available upon reasonable request from the corresponding author.

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
