# Peer review of "Shape- and Element-Sensitive Reconstruction of Periodic Nanostructures with Grazing Incidence X-ray Fluorescence Analysis and Machine Learning"

_nanomaterials, 2021, doi:10.3390/nano11071647_

Round 1

Reviewer 1 Report

What a tiny chance, but it happened that I got to review this paper again. After rereading this paper, I believe my arguments below are still valid.

The paper describes the application of the Bayesian-based method for the 2D profile analysis of line gratings. Overall the paper is well written with enough background and references, but the authors are recommended to consider the following issues seriously before publication.

  1. I do not understand why the discussion on carbon contamination is here. The section appears irrelevant to the topic. First, why not filter out carbon edge signals with energy discrimination on the silicon drift detector? Second, no doubt that GIXRF has the element sensitivity for the carbon sensitivity, but it has no in-plane spatial sensitivity (smaller than beam size) for carbon contamination. The central assumption for the GIXRF analysis here (for a manageable computational speed) is that the grating is 2D in its cross-section. However, the contamination does not have to follow the direction of the gratings.
  2. My major concerns are about the profile model and Bayesian analysis:
    1. Table-II claimed unrealistic posterior uncertainties. For example, h has atomic resolution/uncertainty; w, t_t, and t_g have subatomic resolution/uncertainty. These are impossible values from the experiment data. First, it is questionable whether the GIXRF technique can deliver such a resolution in this particular case. At least there is not standard sample to benchmark the resolution capability here. Second, these posterior uncertainties are strongly biased to the priors, which are the model constraints for the grating profile. The referee is not questioning the excellent fitting with so few parameters for the model but feels these uncertainties do not reflect the possible variations of the physical grating. Overall, the general agreement is good between the SEM and GIXRF reconstruction, but discrepancies are noticeable, especially on the crowns of the gratings. These are probably several nanometers, much more than the atomic and sub-atomic uncertainties in Table II. The SEM images clearly indicate that the underestimated parameter uncertainties are due to strict bounding by the model.
    2. A cross-model study, e.g., with an increased number of profile parameters, may help.
    3. Global optimization is a quite important field in applied mathematics, but Bayesian is not necessarily a global optimization algorithm. It often happens that you may never leave the local basin of traps, even for a stochastic method. But the authors seem to hint they found a global minimum of the reconstruction in section 2.2. This may or may not be true. How to justify that? Multi-start with various initial parameters or priors to help?
    4. Instead of giving the marginal confidence of individual parameters from the diagonal Hessian, the uncertainty profile of the cross-section is of more relevance and interest and should be given. Some of the profile modeling parameters are likely correlated. With the distribution/uncertainty of the parameters (with a correlation matrix) at hand, why not overlay the confidence of the cross-section profile?

Author Response

  1. I do not understand why the discussion on carbon contamination is here. The section appears irrelevant to the topic. First, why not filter out carbon edge signals with energy discrimination on the silicon drift detector? Second, no doubt that GIXRF has the element sensitivity for the carbon sensitivity, but it has no in-plane spatial sensitivity (smaller than beam size) for carbon contamination. The central assumption for the GIXRF analysis here (for a manageable computational speed) is that the grating is 2D in its cross-section. However, the contamination does not have to follow the direction of the gratings.

For a clarification of this remark, we actually do filter carbon signal, because in the recoded fluorescence spectra the carbon fluorescence is a separate peak and after a proper spectral deconvolution it does not influence the nitrogen or oxygen fluorescence signals. Nevertheless, the total measured nitrogen and oxygen fluorescence for small angles is influenced by any surface effects like carbon contamination. Even if we don’t look at the carbon fluorescence the angles distribution of the nitrogen and oxygen signal changes with different surface effects.

We have a microscope image, where a dark line on the grating can be seen. This can be connected to an increase in the carbon signal. Therefore, the assumption that the contamination is smaller than the beam size is not correct and the thick contamination follows the direction of the grating. We tried to make this more clear in the text (line 274).

By analyzing the carbon signal of the spectra for different positions on the sample we can get information about the distribution of the carbon. Additionally by comparing the measured curves for nitrogen and oxygen fluorescence at the position with and without contamination a difference can be observed. Therefore the carbon contamination has an influence on the reconstruction. We assumed that it is interesting to point this out and show that we are sensitive to surface layers, either intentionally deposited or contaminants in the nanometer regime.

  1. My major concerns are about the profile model and Bayesian analysis:

a) Table-II claimed unrealistic posterior uncertainties. For example, h has atomic resolution/uncertainty; w, t_t, and t_g have subatomic resolution/uncertainty. These are impossible values from the experiment data. First, it is questionable whether the GIXRF technique can deliver such a resolution in this particular case. At least there is not standard sample to benchmark the resolution capability here. Second, these posterior uncertainties are strongly biased to the priors, which are the model constraints for the grating profile. The referee is not questioning the excellent fitting with so few parameters for the model but feels these uncertainties do not reflect the possible variations of the physical grating. Overall, the general agreement is good between the SEM and GIXRF reconstruction, but discrepancies are noticeable, especially on the crowns of the gratings. These are probably several nanometers, much more than the atomic and sub-atomic uncertainties in Table II. The SEM images clearly indicate that the underestimated parameter uncertainties are due to strict bounding by the model.

Thank you for this comment, however we want to point out, that the SEM image was made from a witness sample and only provides information about some cross-sections. Due to production variations there can be slight differences between the witness sample and the sample for the GIXRF measurement. The GIXRF reconstruction averages over a large sample area. Therefore, it is not too surprising that there are differences between the SEM image and the GIXRF reconstruction results. However, the comparison does show, that the obtained reconstruction is of reasonable quality.

Secondly, we do not claim that the stated model parameter confidence intervals we provided are to be understood as final uncertainties or error of the parameters. These are confidence intervals resulting from the Hessian matrix of the error function. We don’t provide complete posterior distributions but in future studies we want to provide them by using for example a Markov-Chain-Monte-Carlo simulation for the GIXRF measurements.

b) A cross-model study, e.g., with an increased number of profile parameters, may help.

Thank you for this remark. During the preparation of the presented final reconstruction, several different reconstruction models with more or less parameters (e.g. pitch, height, only one parameter for the oxide thickness or silicon nitride in the grooves) were tested and the model presented in the paper gave the best and most reasonable results.

The computable effort increases with an increase in the number of model parameters. The BO gets significantly slower with more profile parameters. This is a first step to an apply machine learning tool to this problem. In future artificial intelligence can maybe provide better and fester results for more model parameters.

c) Global optimization is a quite important field in applied mathematics, but Bayesian is not necessarily a global optimization algorithm. It often happens that you may never leave the local basin of traps, even for a stochastic method. But the authors seem to hint they found a global minimum of the reconstruction in section 2.2. This may or may not be true. How to justify that? Multi-start with various initial parameters or priors to help?

Of course, after any finite number of iterations, one cannot be certain to have identified the global minimum. But since we have repeated the minimization several times (e.g. in the carbon contamination study or the repeated measurements) and consistently identify the same minimum, we are certain that the presence of other local minima does not hinder the correct parameter reconstruction. It can be proven mathematically that Bayesian optimization with the expected improvement infill criterion as applied here is a global optimization algorithm that is not trapped into local minima. For the applied covariance function (a Matérn 5/2 kernel) a global convergence is guaranteed, if the objective function is everywhere at least one times differentiable (see  E. Vazquez and J. Bect "Convergence properties of the expected improvement algorithm with fixed mean and covariance functions." Journal of Statistical Planning and inference 2010).

Additionally we measured the GIXRF profile at the same position 10 times and applied the reconstruction to each dataset (which is closely comparable to a multi-start with different initial parameters due to the nature of the BO). The calculated standard derivation of the ten reconstruction was in the same regime as the model parameter confidence interval indicating that the calculated error seems to indicate the statistical error of the reconstruction. Because the difference between the 10 measurements are small this is comparable to start the reconstruction 10 times. Additionally in Andrle et al. (2019, 10.1117/12.2526082) we did a benchmark study and compared to BO to Particle Swarm Optimization and started the BO 6 times. This was a slightly different sample, nevertheless the BO always found the same minimum.

d) Instead of giving the marginal confidence of individual parameters from the diagonal Hessian, the uncertainty profile of the cross-section is of more relevance and interest and should be given. Some of the profile modeling parameters are likely correlated. With the distribution/uncertainty of the parameters (with a correlation matrix) at hand, why not overlay the confidence of the cross-section profile?

This is an interesting idea. We tried to visualize the confidence intervals in the cross-section profile but even 3 sigma, were barely visible. In this paper we wanted to show that the BO as a machine learning tool can be utilized for this sample and that we are able to estimate the confidence intervals resulting from the BO reconstruction. In Andrle et al. (2021, arXiv:2102.03189), it was shown for a similar setup and similar samples that there was no correlation. We added a sentence to the paper: “Parameter correlation cannot be identified from the Hessian matrix as we expect from previous investigation based on Bayesian inversion study Andrle et al. (2021, arXiv:2102.03189)” (lines 262-264)

Reviewer 2 Report

The manuscript is of high quality, it is well-written, although the background could be more general to place the work in better context. The results are new and of interest to various researchers as the topic is highly interdisciplinary. Some details need to be clarified and improved before further consideration.

1) The abstract should mention the potential impact of the research work presented in the manuscript. What is the novelty and how the results will influence the subject field?

2) Justification for the selected ML algorithm should be provided. What are the other algorithms that could have been employed? How did the authors ensure that the Bayesian optimization, the global optimization algorithm, will give the best results?

3) How would HR-TEM enhance the work and the reliability of the results instead of the conventional TEM? Do the authors foresee any improvements with more advanced techniques?

4) A more detailed mathematical framework should be added to the manuscript. This is necessary to ensure that the general readership of ‘nanomaterials’ will understand the work. The authors write to a non-specialized audience and therefore a more thorough background should be provided.

5) The use of machine learning for the optimization of materials should be acknowledged and examples briefly mentioned as it is an emerging area (10.1039/D0GC02956D; 10.1039/D0TA05176D).

6) All the assumptions should be systematically listed in the manuscript.

7) Validation, overfitting and the effect of sample numbers should be investigated in depth.

8) The transferability of the results should be discussed. The presented results are specific to the studied system, i.e. case study, or can be applied to other systems?

9) The text in the inset within Figure 4 is not legible, and needs to be enlarged.

10) The results and discussion section has too many references, and therefore it is unclear which are the original results obtained by the authors in this manuscript for the first time, and what percentage of the information is already known in the literature. The authors should specify what the novel results are.

Author Response

1) The abstract should mention the potential impact of the research work presented in the manuscript. What is the novelty and how the results will influence the subject field?

Thank you for pointing this out. We added “This novel approach enables the element sensitive and destruction-free characterization of nanostructures with sub-nm resolution.” (lines 20-21) to the abstract.

2) Justification for the selected ML algorithm should be provided. What are the other algorithms that could have been employed? How did the authors ensure that the Bayesian optimization, the global optimization algorithm, will give the best results?

In Andrle et al. (2019, 10.1117/12.2526082) we compared the BO to the particle swarm optimization for a similar problem and we found that the BO gives better results in a short amount of time. The BO gives a good result in a reasonable amount of time. Because one function evaluation is time consuming (approximately 30s depending on the computer used for the calculation) we need an optimizer that is global and efficient and for our example the BO works best. A clarifying remark was added to the manuscript: “For a detailed benchmark study of the different optimizer methods in comparison to BO see [27] and [41]. For our problem the BO gives good results in a reasonable amount of time [41].” (lines 163-164)

Referee 1 asked a similar questions which we answered in paragraph 2. c).

3) How would HR-TEM enhance the work and the reliability of the results instead of the conventional TEM? Do the authors foresee any improvements with more advanced techniques?

HR-TEM is an interesting method to characterize nanostructure and combined with EDX it can be element sensitive. HR-TEM can be used to get high resolution images of the cross-section but the obtained information is highly localized and may not be representable for the average structure. However, having high quality TEM cross section data at hand is certainly helpful in designing the reconstruction model and for validation as shown. A TEM and EDX measurement can be used as a reference for the reconstruction.

4) A more detailed mathematical framework should be added to the manuscript. This is necessary to ensure that the general readership of ‘nanomaterials’ will understand the work. The authors write to a non-specialized audience and therefore a more thorough background should be provided.

While we appreciate the reviewer’s feedback, we think it is not necessary or may even confuse to provide more equations. We added an additional reference for the Sherman equation: Sherman (1955, 10.1016/0371-1951(55)80041-0) (line 156).

To learn more about the BO approach other publications provide good and understandable explanation (for example Shahriari et al. (2015, 10.1109/JPROC.2015.2494218)). These publications are also referenced in the manuscript. Nevertheless we added some sentences to better explain how the BO works: “The expected improvement EI is crucial to find the best parameters to evaluate the function to find the minimum of function. The expected improvement EI of an unknown function can be calculated with the Gaussian process. The next sampling point is where the EI is at its maximum

x_{n+1} = argmax(EI_n(x))

The EI is high, where we have not evaluate the function or close to the global minimum found.” (lines 156-157)

5) The use of machine learning for the optimization of materials should be acknowledged and examples briefly mentioned as it is an emerging area (10.1039/D0GC02956D; 10.1039/D0TA05176D).

Thank you for this suggestion. A sentence was added to the introduction acknowledging the optimization of materials “Artificial intelligence and machine learning (ML) techniques are studied intensively for a wide range of applications [23-25]. These techniques can help to improve the optimization of different materials.” (lines 74-76)

6) All the assumptions should be systematically listed in the manuscript.

Thank you for this comment. We numbered all relevant assumptions in the manuscript.

Assumption 1: We assume that the posterior distribution is Gaussian. (line 168)

Assumption 2: The Si substrate is oxidized. (line 184)

Assumption 3: Systematic errors not taken into account may increase the final uncertainties. (line 268)

Assumption 4: As far as the low contrast allows, the agreement with respect to line height, sidewall angle, line width and even the oxide layer thicknesses (assuming the bright areas in the SEM to be the oxide, Assumption 4) is reasonable good. (line 337)

7) Validation, overfitting and the effect of sample numbers should be investigated in depth.

Thank you for this remark. We are not sure if we understand this remark correctly but we have studied this model intensively and validated it with SEM and AFM measurements. The method was analyzed additionally for simulated data with slightly different model parameter like pitch or width. In Hönicke et al. (2020, 10.1088/1361-6528/abb557) the method was applied for different samples like 3D boxes. Referee 1 asked a similar questions which we answered in paragraph 2. b).

8) The transferability of the results should be discussed. The presented results are specific to the studied system, i.e. case study, or can be applied to other systems?

In different papers e.g. Hönicke et al. (2020, 10.1088/1361-6528/abb557) it was shown that the described methods works for different samples made of different materials and also for 3D structures like boxes. The novelty of this paper is the estimations of the confidence intervals of the model parameters and the reconstruction of two different elements. The method described here can be used to calculate the confidence intervals for different systems. We added a sentence pointing this out: “This method can also be applied to various systems as described in this paper [22].” (lines 351-352)

9) The text in the inset within Figure 4 is not legible, and needs to be enlarged.

Indeed, the figure was adapted for better readability.

10) The results and discussion section has too many references, and therefore it is unclear which are the original results obtained by the authors in this manuscript for the first time, and what percentage of the information is already known in the literature. The authors should specify what the novel results are.

We revised part of the results and discussion section to make it more clear, which parts are already known and which parts are novel (lines 181, 185-186, 220, 239-240, and 312).

Round 2

Reviewer 1 Report

w.r.t comment 1

A line of carbon contamination precisely along the direction of the grating is an extraordinarily special case. The root assumption of the analysis is that the Maxwell solver assumes a two-dimension cross-section where the 3rd dimension (along the grating) is uniform, at least statistically to the extent of a small variation (which then leads to discussions on line roughness, waviness, etc.). However, contamination generally does not have this luxury for such a 2D cross-section analysis. How does one know that the cross-section of the contaminated grating is identical (statistically) everywhere along the line? The word “contamination” and its section are misleading, as contamination implies unwanted external materials deposited on the sample. Its distribution is often in a random manner. If one has to deposit a special  line along the grating to test the sensitivity of the 2D model/analysis, it cannot be called “contamination.” Using this word is inappropriate here for GIXRF sensitivity analysis. It gives a false illusion that one may use the relative uncertainty analysis of GIXRF for contamination analysis/mapping.  

w.r.t comment 2a and 2b

The uncertainty levels from the posterior distribution are too small compared to a typical/random witness of SEM. Doesn’t the failure of the confidence in covering a typical SEM cross-section obviously imply that the estimator/optimizer is, in fact, stuck in local traps and fails in approaching the desired minimum? Or something wrong with the model? If the uncertainties do not describe the physical sample or give incorrect measures, why spent so much discussion about the uncertainty analysis?  While I agree a thorough examination of the posterior distribution analysis is necessary, this apparent discrepancy must be reconciled one way or another.

Other replies are satisfactory.

Author Response

w.r.t comment 1
A line of carbon contamination precisely along the direction of the grating is an extraordinarily special case. The root assumption of the analysis is that the Maxwell solver assumes a two-dimension cross-section where the 3rd dimension (along the grating) is uniform, at least statistically to the extent of a small variation (which then leads to discussions on line roughness, waviness, etc.). However, contamination generally does not have this luxury for such a 2D cross-section analysis. How does one know that the cross-section of the contaminated grating is identical (statistically) everywhere along the line? The word “contamination” and its section are misleading, as contamination implies unwanted external materials deposited on the sample. Its distribution is often in a random manner. If one has to deposit a special line along the grating to test the sensitivity of the 2D model/analysis, it cannot be called “contamination.” Using this word is inappropriate here for GIXRF sensitivity analysis. It gives a false illusion that one may use the relative uncertainty analysis of GIXRF for contamination analysis/mapping.

We are by no means trying to reconstruct the 2D distribution of the carbon contamination layer. Besides the fact, that the contamination is indeed expected to be inhomogeneous along the grating we expect it to be also more or less randomly distributed on the modeled cross section. What we want to show is merely the fact that our model is not defined in a way that the existing model parameter set allows to model also the contaminated grating. This would result in a twofold problem: On the one hand side, it would mean that the achievable total uncertainties are drastically larger than the so far stated model parameter confidence intervals if an additional “layer”, which is not considered in the model, leads to no observable deviations. Secondly, the fact that unexpected effects not taken into account in the initial model of a given nanostructure is highly likely also when applying the technique to other samples. Then, the fact that discrepancies between model and reality show up by suspicious results for the reconstructed structure parameters is a good indication that a problem exists. Then one can look further to find and resolve the discrepancies.

With respect to the wording, we do not agree with the referee. The carbon layer is not intentionally grown on the sample to test sensitivities as it is a result of other experiments performed on the sample. Therefore, it is a contamination by definition and we have only tried to take advantage of its presence as stated above.

w.r.t comment 2a and 2b
The uncertainty levels from the posterior distribution are too small compared to a typical/random witness of SEM. Doesn’t the failure of the confidence in covering a typical SEM cross-section obviously imply that the estimator/optimizer is, in fact, stuck in local traps and fails in approaching the desired minimum? Or something wrong with the model? If the uncertainties do not describe the physical sample or give incorrect measures, why spent so much discussion about the uncertainty analysis? While I agree a thorough examination of the posterior distribution analysis is necessary, this apparent discrepancy must be reconciled one way or another.

Here we must clarify two things:

First, the witness sample is only a sample which was “treated” in a similar manner as the one analyzed with GIXRF. It is highly likely that slight variations between witness sample and GIXRF sample occur simply due to process variations. For example, if the amount of deposited Si3N4 slightly changes, the observable line height automatically changes too. Since the sample is destroyed after performing the cross-section SEM, we could not use identical samples for this comparison. Consequently, the comparison between witness and GIXRF sample is not to be interpreted in a way that an exact match is to be expected. It only serves as a demonstration that the obtained reconstruction is of reasonable quality (and thus a good indication for not being stuck in a local minimum).

Secondly, the presented model parameter confidence intervals are by no means interpretable as parameter uncertainties. We state this throughout the manuscript and always refer to them as confidence intervals not uncertainties for exactly this reason. In the overall uncertainty budget, things like the model error and other contributions will definitely play a decisive role and total uncertainties will be much higher than the stated confidence intervals. Unfortunately, their assessment is not straightforward and the main target for future research. In order to emphasize this fact even more in the paper, we have adapted the table and created an extra column for the confidence intervals.

For these reasons we technically agree with what the referee criticizes but it is not within the scope of the paper. We believe it is absolutely legitimate to perform the comparison with SEM as we did and interpret it as we did given the aspects stated above. If we would present total uncertainties and an SEM from the same sample and the same spot, the argumentation would be problematic as the referee criticizes.

Reviewer 2 Report

1. Regarding previous comment 1, the authors should specify the actual nanomaterials in the new sentence (lines 20-21).

2. Previous comment 4 should be revisited and at least the relevant literature on the mathematical background provided within the manuscript.

3. Regarding previous comment 5, the authors should invlude the examples.

4. Regarding previous comment 6, the assumptions are still scattered in the manuscript and they should be summarized in a single paragraph to faciltiate understanding the work.

Author Response

  1. Regarding previous comment 1, the authors should specify the actual nanomaterials in the new sentence (lines 20-21).

We added the materials of the nanostructure to the abstract: “This novel approach enables the element sensitive and destruction-free characterization of nanostructures made of silicon nitride and silicon oxide with sub-nm resolution.” (lines 20-21)

  1. Previous comment 4 should be revisited and at least the relevant literature on the mathematical background provided within the manuscript.

The corresponding literature is already cited in the paper but we clarified it even more by adding the sentence: The mathematical background of Bayesian optimization can be found in ref Shahriari et al. (2015, 10.1109/JPROC.2015.2494218).

  1. Regarding previous comment 5, the authors should include the examples.

Here we disagree. It is a very welcomed remark that we should provide acknowledgement for other work being done using these techniques. This we did as discussed. However explicitly stating one or two fields of application does not provide significantly more information and may also be misleading here. We are not writing a review about machine learning but only use it as a tool for our application.

  1. Regarding previous comment 6, the assumptions are still scattered in the manuscript and they should be summarized in a single paragraph to facilitate understanding the work.

We added this paragraph to the paper.

“For the reconstruction we need to make the following assumptions: For the calculation of the confidence intervals we need to assume that the posterior distribution of the model parameters are Gaussian distributed. For the used model we assumed that the Si substrate is oxidized and that any contamination is negligible. For the comparison of the shape of the nanostructure we assumed that the bright areas in the SEM image represent the oxide.”

Round 3

Reviewer 1 Report

The update from the authors is appreciated. 

Reviewer 2 Report

The manuscript improved.